# Wide-Targeted Metabolome Analysis Identifies Potential Biomarkers for Prognosis Prediction of Epithelial Ovarian Cancer

**DOI:** 10.3390/toxins13070461

**Published:** 2021-06-30

**Authors:** Eiji Hishinuma, Muneaki Shimada, Naomi Matsukawa, Daisuke Saigusa, Bin Li, Kei Kudo, Keita Tsuji, Shogo Shigeta, Hideki Tokunaga, Kazuki Kumada, Keigo Komine, Hidekazu Shirota, Yuichi Aoki, Ikuko N. Motoike, Jun Yasuda, Kengo Kinoshita, Masayuki Yamamoto, Seizo Koshiba, Nobuo Yaegashi

**Affiliations:** 1Advanced Research Center for Innovations in Next Generation Medicine, Tohoku University, Sendai 980-8573, Japan; ehishi@ingem.oas.tohoku.ac.jp (E.H.); libin@ingem.oas.tohoku.ac.jp (B.L.); kazuki.kumada@megabank.tohoku.ac.jp (K.K.); keigo.komine.e7@tohoku.ac.jp (K.K.); hidekazu.shirota.e1@tohoku.ac.jp (H.S.); kengo@tohoku.ac.jp (K.K.); masiyamamoto@med.tohoku.ac.jp (M.Y.); koshiba@megabank.tohoku.ac.jp (S.K.); nobuo.yaegashi.c7@tohoku.ac.jp (N.Y.); 2Tohoku Medical Megabank Organization, Tohoku University, Sendai 980-8573, Japan; matsukawa@megabank.tohoku.ac.jp (N.M.); saigusa@tohoku.ac.jp (D.S.); aoki@megabank.tohoku.ac.jp (Y.A.); motoike@megabank.tohoku.ac.jp (I.N.M.); jun-ysauda@miyagi-pho.jp (J.Y.); 3Department of Obstetrics and Gynecology, Graduate School of Medicine, Tohoku University, Sendai 980-8574, Japan; keikudo@med.tohoku.ac.jp (K.K.); keita.tsuji.d8@tohoku.ac.jp (K.T.); shogo.shigeta.a4@tohoku.ac.jp (S.S.); hideki.tokunaga.a1@tohoku.ac.jp (H.T.); 4Medical Biochemistry, Graduate School of Medicine, Tohoku University, Sendai 980-8575, Japan; 5Department of Clinical Oncology, Tohoku University Hospital, Sendai 980-8574, Japan; 6Systems Bioinformatics, Graduate School of Information Sciences, Tohoku University, Sendai 980-8579, Japan; 7Division of Molecular and Cellular Oncology, Miyagi Cancer Center Research Institute, Natori 981-1293, Japan

**Keywords:** metabolomics, epithelial ovarian cancer, biomarker, ratio of kynurenine and tryptophan, uremic toxins

## Abstract

Epithelial ovarian cancer (EOC) is a fatal gynecologic cancer, and its poor prognosis is mainly due to delayed diagnosis. Therefore, biomarker identification and prognosis prediction are crucial in EOC. Altered cell metabolism is a characteristic feature of cancers, and metabolomics reflects an individual’s current phenotype. In particular, plasma metabolome analyses can be useful for biomarker identification. In this study, we analyzed 624 metabolites, including uremic toxins (UTx) in plasma derived from 80 patients with EOC using ultra-high-performance liquid chromatography-tandem mass spectrometry (UHPLC-MS/MS). Compared with the healthy control, we detected 77 significantly increased metabolites and 114 significantly decreased metabolites in EOC patients. Especially, decreased concentrations of lysophosphatidylcholines and phosphatidylcholines and increased concentrations of triglycerides were observed, indicating a metabolic profile characteristic of EOC patients. After calculating the parameters of each metabolic index, we found that higher ratios of kynurenine to tryptophan correlates with worse prognosis in EOC patients. Kynurenine, one of the UTx, can affect the prognosis of EOC. Our results demonstrated that plasma metabolome analysis is useful not only for the diagnosis of EOC, but also for predicting prognosis with the variation of UTx and evaluating response to chemotherapy.

## 1. Introduction

Epithelial ovarian cancer (EOC) is a fatal gynecological cancer, with a 5-year survival rate of approximately 47.7% between 2008 and 2014, this being the 8th lowest among all cancers [1,2,3]. In the USA, EOC is the fifth leading cause of cancer-related deaths among women, and approximately 14,000 women die from EOC yearly [4]. Most cases diagnosed are metastatic (60%). In Japan, the incidence of ovarian cancer has increased by four-fold, and the mortality rate has increased by 2.5-fold in the last 40 years [5]. In 2019, 4733 women died of EOC, making it the ninth leading cause of cancer-related deaths among women [6]. To date, there exists no justifiable screening program for ovarian cancer, and early detection is required to improve the survival outcome of patients with EOC.

The standard treatment for patients with advanced EOC is a multidisciplinary approach with surgery and chemotherapy [7]. The degree of surgical completion and the sensitivity to chemotherapy have a significant impact on survival outcomes. Combination chemotherapy with platinum agents and taxanes is widely used as the first-line treatment for advanced EOC patients. However, the improvement in long-term prognosis for advanced recurrence using conventional therapeutic strategies remain insufficient. Based on clinical research studies, systemic chemotherapy with molecular targeted therapies, such as angiogenesis inhibitors and poly adenosine diphosphate-ribose polymerase (PARP) inhibitors, have been applied as initial treatment for advanced EOC patients and are expected to improve their survival outcomes. Despite the recent progress in genome analysis technology, very few driver genes are capable of drug discovery in EOC. Thus, accurate disease profiling and effective treatment strategies are required to advance the field of personalized medicine in EOC [8].

Omics sciences, including genomics, transcriptomics, proteomics, and metabolomics, have contributed to the development of new medical approaches. In particular, a metabolomics profile reflects the numerous biochemical events occurring in an organism owing to the complex interactions among age, sex, gene transcription, protein expression, physio-pathological conditions (including gut microbiome activity), and environmental effects [9]. Thus, cancer-specific metabolism has recently attracted significant attention as a new disease profile.

By targeting specific metabolites related to the proliferation, progression, and metastasis of cancer cells, some studies have developed therapeutic strategies for various cancers, thereby demonstrating that metabolomics profiling is a valuable strategy for identifying new cancer risk biomarkers [10,11,12,13,14,15]. Recent comparative studies of ovarian cancer patients and healthy female volunteers have identified biomarkers from tissue, plasma, and urine samples [16,17,18]. To date, only CA-125, one of the serum cancer antigens, is recommended as a biomarker for diagnosis or follow-up in daily practice but its sensitivity and specificity are not so high against EOC [19]. Cancer metabolites circulating in the blood are expected to contain valuable information on primary tumors and have thus received increasing attention as novel biomarkers useful in personalized medicine. In particular, plasma metabolomics is a simpler and less invasive method than tissue metabolomics. Plasma metabolomics can also overcome cancer heterogeneity by determining the current phenotype of cancer patients. However, despite numerous studies, the clinical applications of plasma metabolome biomarkers for EOC diagnosis have not yet been demonstrated.

Recently, a wide-targeted metabolome analysis kit, the MxP^®^ Quant 500 kit, that using ultra-high-performance liquid chromatography-tandem mass spectrometry (UHPLC-MS/MS) has been developed and used to quantified more than 500 metabolites including several uremic toxins (UTx) with good reproducibility [20]. To identify metabolites that directly or indirectly regulate the metabolic functions in EOC, we analyzed the plasma metabolites of EOC patients using the MxP^®^ Quant 500 kit. Our findings revealed new biomarkers and metabolites that could serve as targets for the early diagnosis, chemotherapy sensitivity prediction, and prognosis prediction of patients with EOC.

## 2. Results

### 2.1. Sample Information and Data Cleaning

A targeted metabolome analysis was performed on 80 EOC patients and compared their metabolome profile was compared with the Tohoku Medical Megabank Organization (ToMMo) cohorts (control group). We selected plasma metabolome data measured in the same kit of 80 ToMMo cohort samples that matched age, height, weight, and BMI. The characteristics of the two groups are summarized in Table 1. Of the 624 metabolites measured, 300 had missing data exceeding 20% of all samples and were excluded from further analysis. Finally, 324 metabolites, including one alkaloid, 20 amino acids, 21 amino acid-related, 8 bile acids, 7 biogenic amines, 2 carboxylic acids, 1 cresol, 5 fatty acids, two hormones and related, three indoles and their derivatives, two nucleobases and related, one vitamin and cofactor, four acylcarnitines (AC), 13 lysophosphatidylcholines (LysoPC), 69 phosphatidylcholines (PC), 14 sphingomyelines (SM), 10 ceramides (Cer), 3 dihexosylceramides (Hex2Cer), 11 cholesteryl esters (CE), 2 diglycerides (DG), and 125 triglycerides (TG), were analyzed.

### 2.2. Comparison of Metabolomic Profiles

Principal component analysis (PCA) results showed slight separation of the metabolomic profiles of EOC patients and the healthy controls (Figure 1A), whereas orthogonal partial least squares-discriminant analysis (OPLS-DA) results showed strong separation (Figure 1B), suggesting a characteristic metabolome profile for each group. OPLS-DA is an analysis that creates a discrimination model that considers group information, and the separation of the healthy control and the EOC was mainly due to an increase of TGs and a decrease of PCs.

Changes in each metabolite were then compared. Compared with the healthy controls, 79 metabolites, including five amino acids, five amino acid-related, one bile acid, one biogenic amine, one hormone and related, one nucleobase and related, two ACs, two SMs, two Cers, two Hex2Cers, one CE, one DG, and 55 TGs, were significantly increased in EOC patients (Appendix A), whereas 114 metabolites, including 5 amino acids, 4 amino acid related, one biogenic amine, one cresol, 2 fatty acids, one hormone and related, 2 ACs, 13 LysoPCs, 61 PCs, 5 SMs, 5 Cers, 8 CEs, one DG, and 5 TGs, were significantly decreased (Appendix A).

Figure 2 display the heatmap of the top 30 metabolites that showed significant changes in EOC patients. We observed significant increases in cortisol, TG (16:0_40:6), and TG (22:5_34:2) and significant decreases in His, serotonin, C14:1, SM C24:0, LysoPC a C18:0, LysoPC a C20:3, LysoPC a C24:0, and 20 PCs.

### 2.3. Association of Kynurenine and Tryptophan Ratio with Prognosis in EOC Patients

Based on the significant changes in metabolites in EOC patients, we calculated 232 parameters using MetaboINDICATOR^TM^ to identify each metabolic pathway, and the heatmap of the top 30 parameters is shown in Figure 3. Compared with the healthy controls, 22 indicators related to PC, AC, Cer, CE, TG, and SM metabolism, four indicators related to indole metabolism, and four indicators related to amino acid metabolism were significantly changed in EOC patients. These results reflect the fluctuations in the concentration of each metabolite, including significantly increased TGs and decreased PCs.

Moreover, we analyzed the therapeutic effect of chemotherapy and the correlation between prognosis and each metabolic parameter to identify indicators that can predict the treatment prognosis of EOC patients. We classified the prognosis of 80 EOC patients into disease-free group (*n* = 55), tumor-bearing group (*n* = 16), and death group (*n* = 9), and compared the ratio of kynurenine to tryptophan (Kyn/Trp), which reflects indoleamine-2,3-dioxygenase (IDO) activity. Compared with the disease-free group, the cancer-bearing and the death groups had significantly higher ratio of Kyn/Trp. A correlation between patient prognosis and Kyn/Trp was also observed (Figure 4A). On the other hand, the Kyn/Trp value was higher in the group using chemotherapy with poor chemotherapy response (stable disease (SD) and progressive disease (PD), *n* = 3) compared to the group with good chemotherapy response (complete response (CR)and partial response (PR), *n* = 32), but the difference was not significant (Figure 4B).

## 3. Discussion

Metabolome analysis is a promising technique for biomarker discovery. In this study, we analyzed the plasma metabolome profiles of epithelial ovarian cancer (EOC) patients using the MxP^®^ Quant 500 kit and detected 77 significantly increased and 114 significantly decreased metabolites in EOC patients compared with healthy controls.

Plewa et al. measured the metabolites in the serum of ovarian cancer patients and compared them with those in patients with benign ovarian tumors and healthy controls using a wide-targeted metabolome analysis kit (AbsoluteIDQ^®^ p180) [21]. They reported increased levels of kynurenine and decreased levels of histidine, LysoPCs, and PCs in ovarian cancer patients. We similarly found a significantly increased concentration of kynurenine and decreased concentrations of histidine, LysoPC, and PC in the plasma of EOC patients. To date, several metabolomic analyses of the tissues, serum, and plasma of ovarian cancer patients have been conducted. Consistently, these studies reported decreased LysoPC level in ovarian cancer patients, increased PC level in ovarian cancer cells, and heterogeneously distributed PC in ovarian cancer patients. PC can be transformed into LysoPC by phospholipases A1 and A2 and can then be converted into lysophosphatidic acid (LPA) by lysophospholipase D (LPD) [22,23]. LPD level is increased in cancer cells, and its metabolite LPA is involved in cancer cell survival, proliferation, and metastasis through various LPA receptors [24]. We attributed the marked reduction in plasma LysoPC and PC levels in EOC patients to increased LPA production in ovarian cancer, supporting previous reports.

Regarding other lipids, a significant increase in TGs and a significant decrease in CEs were observed in EOC patients [16,18]. Previous studies on metabolomic profiling of ovarian cancer patient samples have not reported consistent results on TGs, and there are only a few studies analyzing CEs. Zeleznik et al. assessed the association between plasma metabolites and the risk of EOC, including both borderline and invasive tumors, and found that circulating TG levels may be a risk biomarker for ovarian cancer, particularly in rapidly fatal tumors [15]. TG is usually stored in adipocytes and peripheral tissues and is the main energy source in the human body [25]. In patients with ovarian cancer, the release of fatty acids from TG is suppressed, and the involvement of TG in the infiltration and metastasis of cancer cells has been reported [26,27]. Braicu et al. revealed that the risk of ovarian cancer positively correlates with the number of carbon atoms and double bonds in fatty acids [28]. Our data also showed increased TG level with unsaturated fatty acids, consistent with previous reports. Thus, this suggests that TG may be involved in EOC progression. In addition, as the levels of TG and CE in the human body are regulated by lecithin–cholesterol acyltransferase (LCAT), the decrease in CE concentration might be due to the reaction associated with increased TG level [29].

The in vivo properties of amino acids play an important role in many biochemical processes, such as differentiation, protein biosynthesis, energy production, and redox reactions. Reactions involving these amino acids are regulated by the metabolic reprogramming of cancer [30,31,32]. In ovarian cancer, decreased levels of methionine, alanine, histidine, tryptophan, lysine, valine, and threonine have been detected [16,18]. Here, we found significantly increased levels of lysine, isoleucine, aspartic acid, glycine, and threonine, and significantly decreased levels of histidine, valine, tryptophan, tyrosine, and glutamine in the plasma of patients with EOC. Similar to previous reports, decreased concentrations of histidine, valine, and tryptophan were observed. This observation has also been reported in many other cancers, including ovarian cancer, which might indicate high amino-acid uptake and consumption in tumor tissues. On the other hand, the changes in other amino acids (i.e., lysine, isoleucine, aspartic acid, glycine, threonine, tyrosine, and glutamine) that we observed were different from the findings of previous studies. We attribute this inconsistency to the differences in the histology and staging of ovarian cancer tissue samples, sample type, and metabolome analysis method used in the studies.

We further analyzed the relationship between prognosis, therapeutic effect, and metabolic index and found a correlation between IDO activity and prognosis of EOC patients. IDO activity is an index represented by the ratio between the concentrations of kynurenine and tryptophan (Kyn/Trp) [33,34,35]. Tryptophan metabolism is also involved in chronic kidney disease (CKD), and kynurenine metabolites are known to be UTx that are increased even in CKD patients [36,37]. In particular, tryptophan catabolism plays a fundamental role in pathways involved in cancer immune resistance. Meanwhile, IDO is a heme-containing enzyme that catalyzes the rate-limiting step of the oxidative metabolism of indole compounds. The conversion of L-tryptophan to N-formyl-L-kynurenine leads to decreased levels of tryptophan and increased levels of kynurenine and its derivatives. IDO is highly expressed in tumor cells, and kynurenine acts as a ligand for the aryl hydrocarbon receptor (AhR), regulates the transcription of various genes, and inhibits T cell function, resulting in the suppression of the immune system [38,39,40].

We found that the Kyn/Trp value was significantly higher in the tumor-bearing and death groups than the disease-free group and was higher in the group with poor therapeutic effect (SD/PD), although the difference was not significant. These results suggest that the kynurenine pathway is enhanced in EOC patients with high IDO activity and may be involved in the deterioration of EOC by suppressing tumor immunity. In addition, significantly decreased serotonin synthesis, 3-indoleacetic acid (3-IAA) synthesis, and sum of indoles, which are alternative pathways of tryptophan metabolism, have been observed in EOC patients, which are consistent with the enhancement of the kynurenine pathway. Thus, IDO inhibitors may be potential therapeutic agents for EOC patients with presumed increased IDO activity.

In conclusion, we performed a targeted plasma metabolome analysis using the MxP^®^ Quant 500 kit and found a metabolite profile characteristic of EOC patients, suggesting that plasma metabolome analysis is a useful technique in the diagnosis of EOC. Furthermore, kynurenine, one of the UTx, was found to be involved in the prognosis of EOC patients, and higher Kyn/Trp correlated with worse prognosis in EOC, which is attributed to increased IDO activity. It was also implied that UTx are closely associated with the prognosis of EOC. Our findings reveal the potential of Kyn/Trp as a biomarker for prognosis prediction in EOC patients and a target for EOC treatment.

## 4. Materials and Methods

### 4.1. Study Design and Sample Collection

A total of 80 patients with the International Federation of Gynecology and Obstetrics (FIGO) stages I–IV EOC with histologically confirmed diagnosis, who were treated between November 2017 and November 2019 at the Department of Gynecology, Tohoku University Hospital, were eligible for this study.

Before the initial treatment for EOC, biological samples, including plasma, were collected as a clinical biobank project of the Personalized Medicine Center of the Tohoku University Hospital and stored at the clinical biobank of the Advanced Research Center for Innovations in Next Generation Medicine (INGEM). The clinical biobank project of the Personalized Medicine Center, Tohoku University Hospital, was approved by the ethical committee of the Tohoku University School of Medicine (approval number: 2017-1-346, approval date: 8 August 2017). All patients provided written informed consent, and this study was conducted in accordance with the principles of the Declaration of Helsinki.

For the collection and storage of the plasma samples from EOC patients, we conducted the similar protocol for the ToMMo cohort [41,42,43]. Briefly, blood samples were collected on tubes containing EDTA- 2Na that were immediately inverted 10 times, stored at 4 °C, and transported to our biobank laboratory using refrigerated containers with an ice box. The transported tubes were centrifuged at 2330× *g* for 10 min at 4 °C. The plasma fraction was aliquoted into 1 mL matrix 2D barcoded storage tubes (BC30661, Thermo Fisher Scientific, Waltham, MA, USA) by manual processing and stored at −80 °C until further use.

### 4.2. Materials

The following reagents were used in this study: pyridine (C_5_H_5_N; Tokyo Chemical Industry, Tokyo, Japan); ethanol (C_2_H_5_OH; Nacalai Tesque, Kyoto, Japan); phenyl isothiocyanate (C_6_H_5_NCS), formic acid (HCOOH), ammonium acetate (CH_3_COONH_4_; all from Wako Pure Chemical Industries, Osaka, Japan); and pooled normal human plasma as a global quality control (gQC; Innovative Research, Novi, MI, USA). All other chemicals and reagents used were of the highest quality commercially available.

### 4.3. Sample Preparation

Targeted metabolomic analysis was performed using the Biocrates MxP^®^ Quant 500 kit (Biocrates Life Science AG, Innsbruck, Austria) with an ultra-performance liquid chromatograph (ACQUITY UPLC H-Class, Waters Corporation, Milford, MA, USA) connected to a triple quadrupole mass spectrometer (MS; Xevo TQ-S, Waters Corporation) as previously described [44]. The blank solution (10 μL), calibration standard solutions, quality control solutions, and plasma samples were added to the predetermined wells of a 96-well plate. Then, the plate was dried and 5% phenyl isothiocyanate was added to all wells for the derivatization of amino acids and biogenic amines in the samples. The derivatized samples were dried using a pressure manifold (Positive Pressure-96 Processor, Waters Corporation), eluted with 5 mmol/L ammonium acetate in methanol, and diluted to two-fold with water for liquid chromatography (LC)-MS/MS and 50-fold with the flow injection analysis (FIA) mobile phase (FIA mobile phase additive + 290 mL methanol) for FIA-MS/MS. The LC column was an MxP^®^ Quant 500 kit system column system (Biocrates Life Science AG) at 50 °C with the gradient elution of mobile phases A: 0.2% formic acid in water and B: 0.2% formic acid in acetonitrile. The total run time of LC mode was 5.8 min for each analysis at positive ion mode or negative ion mode, respectively. The total run time was 3.0 min for each of three FIA modes. The optimal parameters of ionization, ion transfer voltages, ion transfer temperatures, and the detection of *m/z* pair of precursor and product ion at multiple reaction monitoring (MRM) mode were automatically set using the method in the MxP^®^ Quant 500 kit. The five μL, 15 μL, or 20 μL was injected to the system for LC positive ion mode, LC negative ion mode, or FIA mode, respectively. Metabolite concentrations were calculated using the exported raw file in the MetIDQ^TM^ Oxygen software (Biocrates Life Science AG).

### 4.4. Data Management and Statistical Analysis

The metabolome data of EOC patients were compared to those of the ToMMo cohort matched for age, height, weight, and BMI [45,46]. Data normalization was based on the median of gQC of each plate (four replicates/plate). Metabolites above the limit of detection (LOD) of at least 80% of all samples were used for statistical analysis. Multivariate analysis, PCA, and OPLS-DA were performed using the SIMCA-P v16 software (Umetrics, Umeå, Sweden) and Metaboanalyst 5.0. *P*-values were calculated using the Wilcoxon rank-sum test with the Shapiro-Wilk test, and 232 metabolic parameters (sum or ratio) were calculated using MetaboINDICATOR^TM^ (Biocrates Life Science AG). Differences with *p* < 0.05 were considered statistically significant.

## Figures and Tables

**Figure 1 toxins-13-00461-f001:**
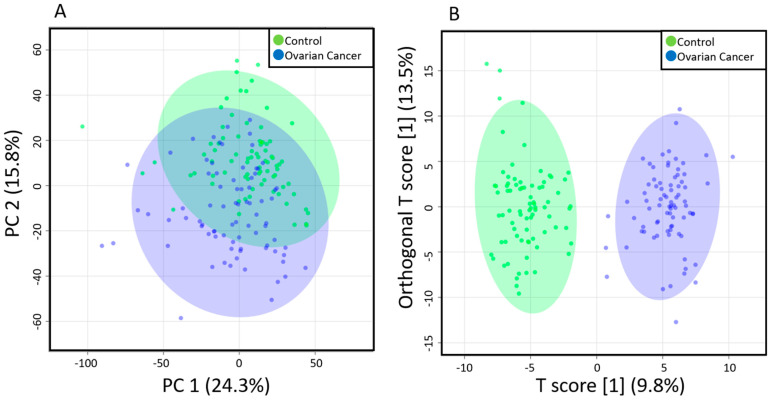
Multivariate analysis results of the plasma metabolites of EOC patients and healthy controls. (**A**) Principal components analysis (PCA) results showing a slight separation between the healthy controls (blue) and EOC patients (green); (**B**) Orthogonal partial least squares-discriminant analysis (OPLS-DA) results showing a strong separation between the healthy controls (blue) and EOC patients (green). Each point in the plot corresponds to one plasma sample.

**Figure 2 toxins-13-00461-f002:**
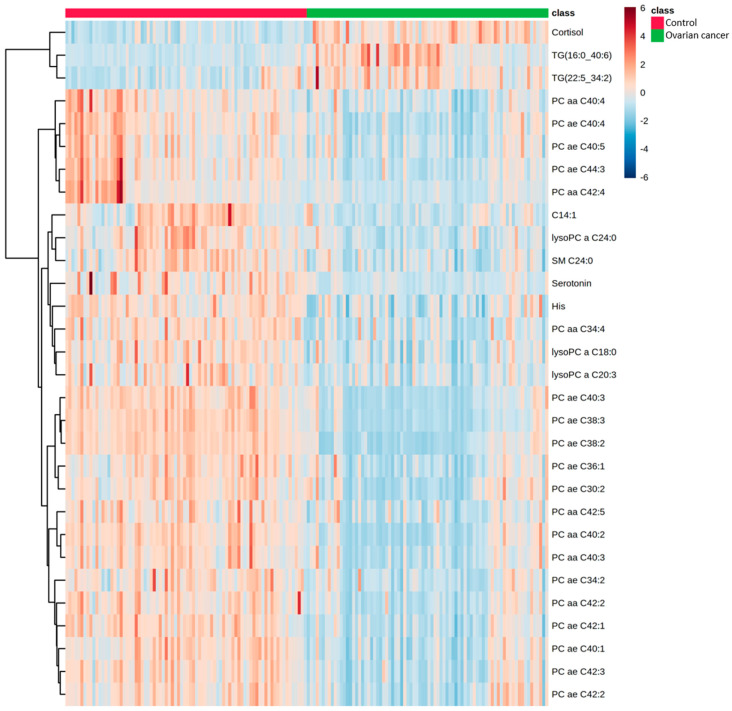
Hierarchical clustering of metabolites in EOC patients and healthy controls. Horizontal columns represent the concentration of each metabolite that displays distinct metabolic patterns between EOC patients and healthy controls. Blue bars indicate decreased level in EOC patients, whereas red bars indicate increased level in EOC patients. The dendrogram on the left is codirected based on the metabolite concentration profiles.

**Figure 3 toxins-13-00461-f003:**
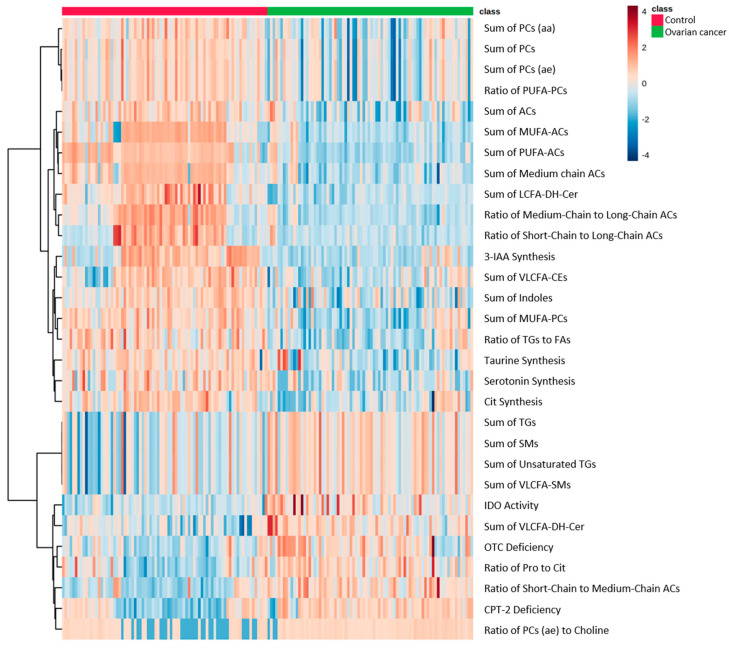
Heatmap showing MetaboINDICATOR hierarchical clustering of the metabolomic profiles of EOC patients and healthy controls. Horizontal columns represent the intensity of each parameter displaying distinct metabolic patterns in EOC patients and healthy controls as calculated by MetaboINDICATOR. Blue bars indicate decreased level in EOC patients, whereas red bars indicate increased level in EOC patients. The dendrogram on the left is codirected based on the metabolic parameter profiles.

**Figure 4 toxins-13-00461-f004:**
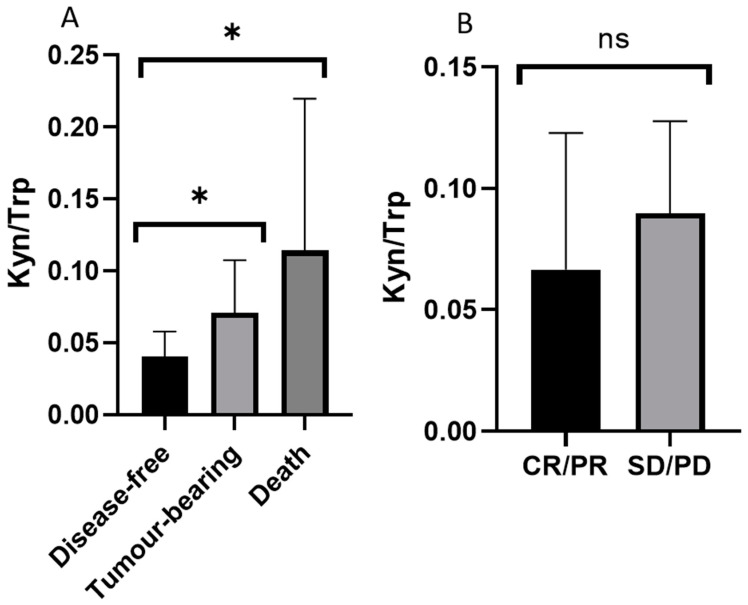
Relationships between the kynurenine-tryptophan ratio (Kyn/Trp) and the prognosis of and chemotherapy response in EOC patients. (**A**) An inverse relationship between Kyn/Trp and patient prognosis was detected (* *p* < 0.05 vs. disease-free control group analyzed by Kruskal-Wallis post-hoc test). (**B**) No significant relationship between Kyn/Trp and chemotherapy effects was observed. CR: complete response; PR: partial response; SD: stable disease; PD: progressive disease.

**Table 1 toxins-13-00461-t001:** Characteristics of the healthy controls and epithelial ovarian cancer patients.

Characteristics	Epithelial Ovarian Cancer Patients (*n* = 80)	Healthy Controls (*n* = 80)	*p*
Age (years, mean ± SD)	58 ± 13	59 ± 12	0.32
Height (cm, mean ± SD)	157 ± 5	156 ± 6	0.35
Weight (kg, mean ± SD)	54 ± 9	55 ± 12	0.89
BMI (kg/m^2^, mean ± SD)	22 ± 3	23 ± 4	0.43
FIGO stage, *n* (%)			
I	26 (32.50)		
II	6 (7.50)		
III	34 (42.50)		
IV	13 (16.25)		
NA	1 (1.25)		
Histopathological type, *n* (%)			
High-grade serous	35 (43.75)		
Low-grade serous	3 (3.75)		
Clear cell	16 (20.00)		
Endometrioid	11 (13.75)		
Mucinous	8 (10.00)		
Others	7 (8.75)		

*p*-values were calculated using the Mann-Whitney test. NA represents not available.

## Data Availability

Data are available upon request, please contact the contributing author.

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
