# Peer review of "Wide-Targeted Metabolome Analysis Identifies Potential Biomarkers for Prognosis Prediction of Epithelial Ovarian Cancer"

_toxins, 2021, doi:10.3390/toxins13070461_

Round 1
Reviewer 1 Report
Review “Wide-targeted metabolome analysis identifies potential biomarkers for prognosis prediction of epithelial ovarian cancer,” submitted to Toxins
Summary: The submission describes a study using plasma metabolomics to look for differences in metabolites between a set of ovarian cancer patients and cancer-free controls matched for age and BMI. A commercially available kit is used for sample processing. Some metabolites are found to be elevated in the cancer cases, while others are lowered, relative to the controls. One particular ratio, of kynurenine/tryptophan is suggested to correlate with worse prognosis.
Comments:
The authors report differences between case and control with modest significance. As a proof of concept study with a relatively small sample set, I am persuaded that the methods and results, though not likely to be highly impactful, are worth reporting. I would prefer to see the analysis disaggregated by cancer type or stage, but the statistical power of the study (with n = 80 individuals) will not make that possible. The authors are encouraged to extend the study to include more individuals.
Major concerns include:
(1) By one mechanism of multivariate analysis (PCA, shown in Figure 1 A), the difference between cancer and non-cancer samples is only slight, while the other method (OPLS-DA, shown in Figure 1B) shows a clear difference. The authors should explain the reason for this difference.
(2) The authors have not included a detailed enough description of how the mass spectrometry analysis was performed to enable another researcher to reproduce their experiments. More detail should be provided.
(3) Sampling and storage conditions can have significant effects on the results of any ‘omics analysis in which a case and control are compared. The authors mention that samples from cancer patients were collected using a “similar protocol for the ToMMo cohort.” Can the authors provide any evidence that there was no systematic bias introduced during sampling and storage?
Minor concerns include:
(1) Figure 1 is difficult to read. The text is small and blurry. Please remake the figure to make it more legible.
(2) Standard deviations are typically not reliable to more than 1 significant figure. The data in Table 1 report averages and standard deviations with three or four significant figures. The table should be remade with the standard deviations rounded to one significant figure and the data reported to the same decimal place as the standard deviation.
(3) The term “cohort” is confusing when used to describe the set of individuals without cancer. I would suggest that the authors consider changing “cohort” to “control.”
(4) I was not able to find whether the origin of the sample was known to the researcher at the time of analysis. Ideally, the researcher would have been unaware of the origin of the sample, to prevent the introduction of unconscious bias. The authors should comment on whether or not this “blinding” was done.
Author Response
Reviewer 1
Summary: The submission describes a study using plasma metabolomics to look for differences in metabolites between a set of ovarian cancer patients and cancer-free controls matched for age and BMI. A commercially available kit is used for sample processing. Some metabolites are found to be elevated in the cancer cases, while others are lowered, relative to the controls. One particular ratio, of kynurenine/tryptophan is suggested to correlate with worse prognosis.
Comments:
The authors report differences between case and control with modest significance. As a proof of concept study with a relatively small sample set, I am persuaded that the methods and results, though not likely to be highly impactful, are worth reporting. I would prefer to see the analysis disaggregated by cancer type or stage, but the statistical power of the study (with n = 80 individuals) will not make that possible. The authors are encouraged to extend the study to include more individuals.
Major concerns include:
Comment 1: By one mechanism of multivariate analysis (PCA, shown in Figure 1 A), the difference between cancer and non-cancer samples is only slight, while the other method (OPLS-DA, shown in Figure 1B) shows a clear difference. The authors should explain the reason for this difference.
Response: In accordance with the Reviewer's suggestion, we have amended Section 2.2. as follows:
Page 3, Line 120: Principal component analysis (PCA) results showed slight separation of the metabolomic profiles of EOC patients and the healthy controlhorts (Figure 1A), whereas orthogonal partial least squares-discriminant analysis (OPLS-DA) results showed strong separation (Figure 1B), suggesting a characteristic metabolome profile for each group. In the case of OPLS-DA, it is an analysis that creates a discrimination model that considers group information, and the separation of the healthy control and the EOC was mainly due to an increase of TGs and a decrease of PCs.
Comment 2: The authors have not included a detailed enough description of how the mass spectrometry analysis was performed to enable another researcher to reproduce their experiments. More detail should be provided.
Response: In accordance with the Reviewer's suggestion, we have amended Section 4.3. as follows:
Page 10, Line 303: The LC column was an MxP® Quant 500 kit system column system (Biocrates Life Science AG) at 50 oC with the gradient elution of mobile phases A: 0.2% formic acid in water and B: 0.2% formic acid in acetonitrile, and 0.2% formic acid in water and acetonitrile were used as mobile phases A and B, respectively. The total run time of LC mode was 5.8 min for each analysis at positive ion mode or negative ion mode, respectively. The total run time was 3.0 min for each of three FIA modes. The optimal parameters of ionization, ion transfer voltages, ion transfer temperatures, and the detection of m/z pair of precursor and product ion at multiple reaction monitoring (MRM) mode were automatically set using the method in the MxP® Quant 500 kit. The five μL, 15 μL, or 20 μL was injected to the system for LC positive ion mode, LC negative ion mode, or FIA mode, respectively. Metabolite concentrations were calculated using the exported raw file in the MetIDQTM Oxygen software (Biocrates Life Science AG).
Comment 3: Sampling and storage conditions can have significant effects on the results of any ‘omics analysis in which a case and control are compared. The authors mention that samples from cancer patients were collected using a “similar protocol for the ToMMo cohort.” Can the authors provide any evidence that there was no systematic bias introduced during sampling and storage?
Response: We thank the Reviewer for these insightful comments. It is a pity that we cannot provide any data to prove that there was no systematic bias introduced during sampling and storage now. However, all of the collected cancer samples are stored in the same facility as ToMMo and are processed by almost the same protocol, therefore, the systematic bias is considered to be very small. In addition, some kind of benign-tumor samples and other cancer types of samples were analyzed and the data showed no significant difference in metabolite profile from the cohort sample. Thus, the reflect changes in metabolites in our data could be considered to be due to EOC.
Minor concerns include:
Comment 1: Figure 1 is difficult to read. The text is small and blurry. Please remake the figure to make it more legible.
Response: In accordance with the Reviewer's suggestion, we have remaked figure 1.
Comment 2: Standard deviations are typically not reliable to more than 1 significant figure. The data in Table 1 report averages and standard deviations with three or four significant figures. The table should be remade with the standard deviations rounded to one significant figure and the data reported to the same decimal place as the standard deviation.
Response: In accordance with the Reviewer's suggestion, we have remaked Table 1.
Comment 3: The term “cohort” is confusing when used to describe the set of individuals without cancer. I would suggest that the authors consider changing “cohort” to “control.”
Response: In accordance with the Reviewer's suggestion, we have changed “cohort” to “control.”
Comment 4: I was not able to find whether the origin of the sample was known to the researcher at the time of analysis. Ideally, the researcher would have been unaware of the origin of the sample, to prevent the introduction of unconscious bias. The authors should comment on whether or not this “blinding” was done.
Response: We thank the Reviewer for these insightful comments. Same as the comment, when measuring a sample, the analyst is not informed which sample corresponds to which cancer type since the anonymization and ID conversion are performed before the sample is provided from the biobank.
Reviewer 2 Report
The authors have made a brilliant work on this manuscript. I would only add the following minor suggestions:
Line 34-35: “In 2019, 4,733 women died of EOC, making it the 7th leading cause of cancer-related deaths among women.” Kindly consider providing a reference to this remark.
Line 45: “PARP inhibitors”. Kindly consider providing the full name before using the abbreviation of poly adenosine diphosphate-ribose polymerase in the text.
Line 87-95: Kindy elaborate on the numbers here. 624 metabolites were initially measured and 306 were excluded due to missing data – that would result in 318 metabolites being included in the final analysis (324 being the stated number which adds up in the list provided). I assume that the 6 extra values analyzed refer to derivatives and/or related metabolites?
Figure 1: Kindly make an effort to provide an image of better analysis.
Author Response
Reviewer 2
The authors have made a brilliant work on this manuscript. I would only add the following minor suggestions:
Comment 1: Line 34-35: “In 2019, 4,733 women died of EOC, making it the 7th leading cause of cancer-related deaths among women.” Kindly consider providing a reference to this remark.
Response: As reviewer’s kind suggestion, we added the reference as following; Cancer Statistics in Japan -2021. Edited by Foundation for promotion of cancer research as reference [6]. In addition, we revised the statement as following; In 2019, 4,733 women died of EOC, making it the 9th leading cause of cancer-related deaths among women. (Line 55-56)
Comment 2: Line 45: “PARP inhibitors”. Kindly consider providing the full name before using the abbreviation of poly adenosine diphosphate-ribose polymerase in the text.
Response: In accordance with the Reviewer's suggestion, we have amended Section 1 as follows:
Page 2, Line 66: Based on clinical research studies, systemic chemotherapy with molecular targeted therapies, such as angiogenesis inhibitors and poly adenosine diphosphate-ribose polymerase (PARP) inhibitors, have been applied as initial treatment for advanced EOC patients and are expected to improve their survival outcomes.
Comment 3: Line 87-95: Kindy elaborate on the numbers here. 624 metabolites were initially measured and 306 were excluded due to missing data – that would result in 318 metabolites being included in the final analysis (324 being the stated number which adds up in the list provided). I assume that the 6 extra values analyzed refer to derivatives and/or related metabolites?
Response: In accordance with the Reviewer's suggestion, we have amended Section 2.1. as follows:
Page 3, Line 109: Of the 624 metabolites measured, 300 had missing data exceeding 20% of all samples and were excluded from further analysis.
Comment 4: Figure 1: Kindly make an effort to provide an image of better analysis.
Response: In accordance with the Reviewer's suggestion, we have remaked figure 1.